# Effect of rotating providers on chest compression performance during simulated neonatal cardiopulmonary resuscitation

**Tavleen Sandhu**[1]*, **Edgardo G. Szyld**[1], **Michael P. Anderson**[2], **Birju A. Shah**[1]

**1** Department of Pediatrics, Section of Neonatal-Perinatal Medicine, The University of Oklahoma Health Sciences Centre, Oklahoma City, OK, United States of America, **2** Department of Biostatistics and Epidemiology, The University of Oklahoma Health Sciences Centre., Oklahoma City, OK, United States of America

* tavleen-sandhu@ouhsc.edu

**Data Availability Statement:** All relevant data are within the paper and its Supporting Information files.

## Abstract

### Objective

Simulation studies in adults and pediatrics demonstrate improvement in chest compression (CCs) quality as providers rotate every two minutes. There is paucity of studies in neonates on this matter. This study hypothesized that frequent rotation while performing CCs improves provider performance and decreases fatigue.

### Study design

Prospective randomized, observational crossover study where 51 providers performed 3:1 compression-ventilation CPR as a pair on a term manikin. Participants performed CCs as part of 3 simulation models, rotating every 3, 5 and 10 minutes. Data on various CC metrics were collected. Participant vitals were recorded at multiple points during the simulation and participants reported their level of fatigue at completion of simulation.

### Results

No statistically significant difference was seen in any of the compression metrics. However, differences in the providers' fatigue scores were statistically significant.

### Conclusion

CC performance metrics did not differ significantly, however, providers' vital signs and self-reported fatigue scores significantly increased with longer CC durations.

## Introduction

Numerous simulation studies describe the need for frequent rotation of healthcare providers while performing chest compressions (CCs) on both adults and pediatrics. These studies

**Funding:** The authors received no specific funding for this work.

**Competing interests:** The authors have declared that no competing interests exist.

unanimously support that the quality of CCs improve when providers rotate every 2 minutes, and hence the recommended frequency of rotation in the adult basic life support (BLS) and paediatric advanced life support (PALS) guidelines [1,2]. A more recent adult simulation study showed that rotation after one minute of continuous CCs reduces rescuer fatigue and improves quality of compressions, compared to the standard practice of rotation every two minutes [3]. Of note, current PALS guidelines were extrapolated from the adult BLS guidelines. To date, few simulation studies have been conducted on paediatric resuscitation. One such study hypothesized that compression quality, and the work needed to accomplish them would differ in a child versus an adult manikin model, given the difference in body habitus [4]. However, to their surprise, CC quality deteriorated similarly in both child and adult manikin models, and the peak work per compression cycle was comparable.

CCs are an infrequent intervention during neonatal resuscitation in the delivery room, and are required by approximately 0.1% of term infants and in up to 15% of preterm infants [5,6]. However, compressions remain an essential component of advanced neonatal resuscitation whenever the neonate's heart rate falls below 60 beats per minute (bpm). As per the current Neonatal Resuscitation Program (NRP) and European Resuscitation Council (ERC) recommendations, CCs should be started if the heart rate remains below 60 bpm in spite of at least 30 seconds of effective ventilation, using a 3:1 CCs to ventilation (CV) ratio, providing 90 CCs and 30 ventilations per minute [7,8]. At present, none of these guidelines offer specific recommendations for provider rotation during neonatal resuscitation, despite recent studies stressing the importance of provider rotation [9–12]. In addition, data on the role of human factors and team science in neonatal resuscitation are scarce. A few recent simulator-based neonatal resuscitation studies on term manikins demonstrate fatigue development and CC quality deterioration over time during prolonged resuscitations [9,10]. These studies demonstrate significant CCs quality reduction as early as 3 minutes of provider performance. These studies support the need for further research assessing the change in CC performance quality when providers rotate frequently. To date, no work has been done in neonates to determine the quality of CCs while comparing frequent rotation with the current standard practice of no rotation, or rotation when fatigued. This study, hypothesized that frequent rotation would improve the providers' performance and lowers their level of fatigue during a prolonged neonatal resuscitation. Its ultimate goal is to incorporate the practice of provider rotation during neonatal resuscitation in the NRP and ERC guidelines.

## Materials and methods

### Subjects

This study was conducted at a large academic perinatal referral centre. A targeted convenient sample size of fifty participants was selected. NRP-certified attendings, fellows, neonatal nurse practitioners, and transport team nurses that routinely attend at risk deliveries were eligible to participate. Providers with a medical condition that restricted them from performing a strenuous activity like prolonged CCs, including a poorly controlled cardiac, respiratory, musculoskeletal and/or neurologic ailment, were considered ineligible. Participation was voluntary, and all participants meeting the inclusion criteria were asked to provide written, informed consent. The study was approved by the Institutional Review Board at the University of Oklahoma (IRB# 9401).

### Protocol

This was a prospective randomized, provider crossover study using a manikin-based simulation procedure assessing the change in providers' CC performance as they rotated frequently

while performing cardiopulmonary resuscitation (CPR) per the recommended 3:1 CV ratio. Providers were paired and asked to perform the following three simulations: A) three rounds of CCs rotating every 3 minutes for a total of 18 minutes; B) two rounds of CCs rotating every 5 minutes for a total of 20 minutes and C) one full round of CCs rotating every 10 minutes or until the provider was too fatigued to perform for maximum total of 20 minutes. Our rationale for choosing a 3-minute rotation schedule was based on the results of the two studies demonstrating a reduction in CCs efficiency after 3 minutes of CCs [9,10]. The 10-minute rotation schedule was based on the resuscitation guidelines at the time of study design, advising that it is reasonable to consider stopping resuscitation if heart rate remains undetectable for 10 minutes [7,8]. Lastly, the 5-minute rotation schedule was chosen as an intermediate duration between 3 and 10 minutes.

Each provider had a different partner for each of the three sessions, tallying to a total of 75 simulation sessions. The RedCap® computer randomization software was used to generate the sequence in which each provider performed the three simulations, as shown in Table 1. This was done to reduce any possibility of CCs performance improvement due to practice. A manual log of completed sessions was kept and no two providers were paired with each other for any subsequent simulations. Each participant was required to perform all three simulations. The three simulations were performed at least 24 hours apart to minimize any residual fatigue from the prior performance. To further eliminate residual fatigue, end of shift participation was not permitted. A term Laerdal Resusci® Baby QCPR manikin (Stavanger, Norway), connected to a Laerdal SimPad PLUS with SkillReporter® 206–30001 (Stavanger, Norway) device was used in this study, shown in Fig 1. The device has the capability to give audio-visual feedback during practice session, and record performances of the paired providers.

The study was conducted in a simulation education room located close to the NICU, for convenience. Sessions were performed on an activated Drager® warmer (Lübeck, Germany) set to deliver heat at 100% to simulate a real resuscitation scenario. Providers were given an option to use a step stool and allowed to adjust the stool height as needed. CCs were administered as per the current NRP and ERC guidelines [7,8]. The providers were asked to perform CCs from the head of the bed, using the two-thumb encircling technique, positioning thumbs on the lower third of the sternum, compressing to a depth of at least one-third of the antero-posterior (1/3 AP) diameter of the manikin's chest, allowing full chest recoil between compressions without losing contact with the chest wall. The manikin was pre-intubated and ventilation was provided by a member of the research team while standing to the right of the manikin, using the t-piece resuscitator, coordinating breath delivery using the 3:1 CV ratio, targeting 90 CCs and 30 breaths a minute. Prior to conducting the simulation, all providers were given a chance to practice compressions to familiarize themselves with the chest compliance and the anatomy of the manikin. Providers had their vitals checked at baseline and after each round of CCs. They were then provided a rest period until their next round of CCs.

**Table 1. Six variations in the sequence in which simulations were performed.**

| Sequence | First session | Second session | Third session |
|---|---|---|---|
| 1 | Rotation every 3 minutes | Rotation every 5 minutes | Rotation every 10 minutes |
| 2 | Rotation every 3 minutes | Rotation every 10 minutes | Rotation every 5 minutes |
| 3 | Rotation every 5 minutes | Rotation every 3 minutes | Rotation every 10 minutes |
| 4 | Rotation every 5 minutes | Rotation every 10 minutes | Rotation every 3 minutes |
| 5 | Rotation every 10 minutes | Rotation every 3 minutes | Rotation every 5 minutes |
| 6 | Rotation every 10 minutes | Rotation every 5 minutes | Rotation every 3 minutes |

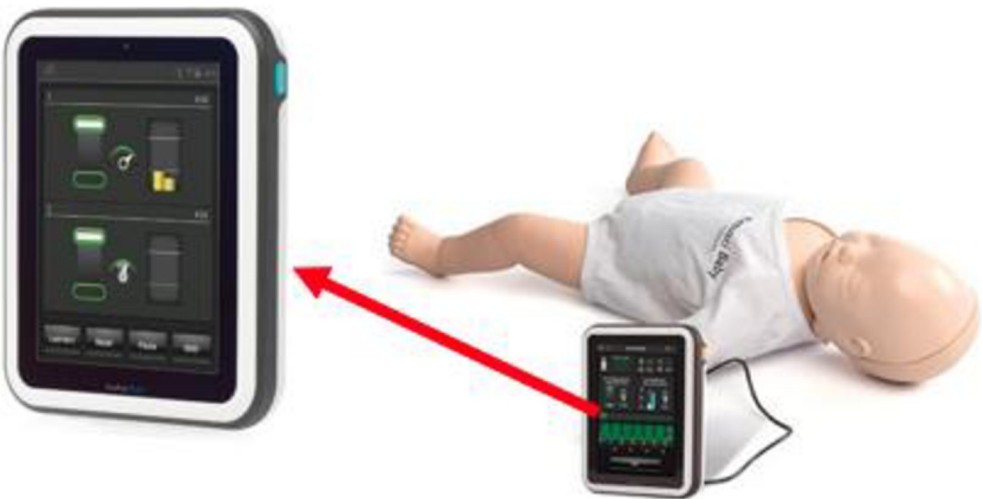

**Fig 1. Term Laerdal Resusci® Baby QCPR manikin (Stavanger, Norway) and Laerdal SimPad PLUS with SkillReporter® 206–30001 (Stavanger, Norway).**

Neither participants, nor the research team were blinded to the objectives of the study given the restricted pool of providers and their familiarity with the study.

## Measurements

Data, including percent CCs with adequate rate, percent CCs with adequate depth, percent CCs with complete recoil, percent CCs with correct thumb positioning, percent overall performance and duration of provider rotation pauses, were obtained from the Laerdal SimPad PLUS with SkillReporter® device. The device was set to assess compression performance only. Thus, no ventilation metrics were obtained. Providers had no access to their performance tracing on the SimPad PLUS with SkillReporter® and they received no live feedback about their performance once the session commenced. A member of the research team recorded baseline heart rate, blood pressure and oxygen saturation before the simulation began. Repeat values were obtained at the end of each round of CCs. Vital signs were obtained using a Welch Allyn 4700–60® monitor (Skaneateles Falls, NY). All providers completed an anonymous questionnaire inquiring about demographic details including gender, age, weight, height, smoking and physical activity status. At the completion of each simulation, providers anonymously rated their level of fatigue on a Likert scale of 1-to-10, where a score of 1 represented no fatigue and a score of 10 represented extreme fatigue. Questionnaire can be found in S1 Fig, it was not validated. Anonymity was preferred in order to ensure honesty and substantial enrolment.

## Data analysis

Descriptive statistics were calculated for all demographic and study-specific measures, such as vitals, fatigue scores and performance variables. Categorical variables were tabulated as counts and percentages. Continuous variables were assessed for normality using the Shapiro-Wilk test and calculated as Mean (SD) or Median ($25^{th}$-$75^{th}$%), as appropriate. Comparisons of performance variables were made between the three groups using the Kruskal-Wallis test. Differences in the subject vitals between pre- and post- vital sign measures were calculated, assessed for normality and compared between the three groups using the Kruskal-Wallis test. Since only two groups had additional follow-up periods, those differences were compared using the

Wilcoxon-Mann-Whitney test. Comparisons in the change from baseline to follow-up within each group were done using repeated measures ANOVA for the 3-minute rotation and a paired t-test for the 5- and 10-minutes rotations. A P-value of <0.05 was considered statistically significant.

## Results

Fifty-one neonatal healthcare professionals were enrolled in this study as shown in Fig 2. Simulations were conducted between April and October 2019. All providers consented to participate. Participant demographics and characteristics are described in Tables 2 and 3. One participant did not disclose his/her weight. None reported smoking. We maintained a minimum of 24 hours gap between simulations for all participants. At the time of data entry, it was discovered that two pairs of providers were unintentionally paired together for one of their subsequent simulations, and another provider inadvertently performed the simulation of every three-minute rotation twice, and missed the every five-minute rotation. Providers that were accidentally paired together had a gap of 10 and 25 days from their prior paired session. With the error in the pairing process, the total number of possible simulations summed to be less than the anticipated 75. Hence, one additional participant was enrolled for a total of 51 participants and 76 paired simulations.

Data on percent CCs with adequate rate, percent CCs deep enough, percent CCs fully released, percent CCs with correct thumb positioning and duration of pauses as providers rotated, were collectively assessed for all three simulation arms and then compared. No statistical significance was found for any of these compression metrics as shown in Fig 3. On a closer

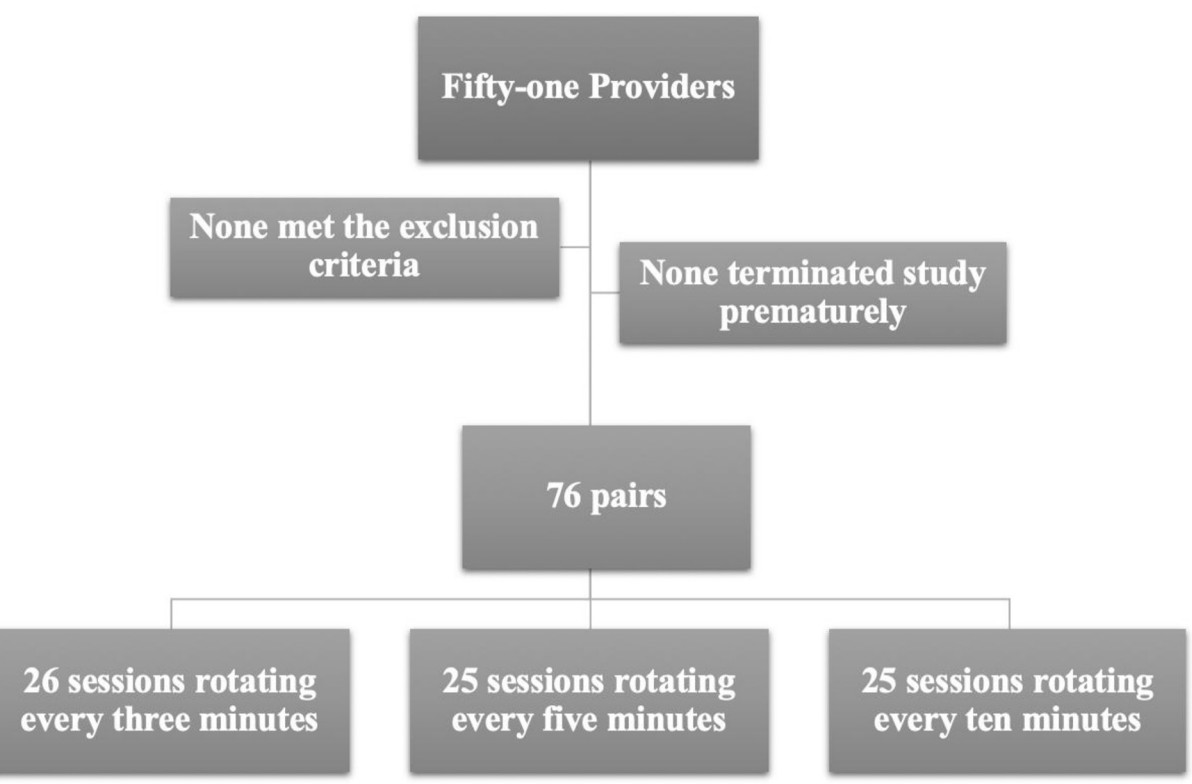

**Fig 2. CONSORT algorithm illustrating study enrolment and reporting number of sessions under each study arm.** Note that, one additional provider (with a total of 51 as oppose to initially determined 50 providers) was enrolled due to error in the pairing process.

**Table 2. Participant population composition.**

| Variables | Frequency (%) |
|---|---|
| Female | 42 (82%) |
| Attending | 15 (29%) |
| Fellows | 6 (12%) |
| Nurse Practitioners | 25 (49%) |
| Neo-flight Nurses | 5 (10%) |

observation, it was found that 5 providers performed very shallow CCs consistently, skewing the performance metrics for the pair. Data were re-analysed after removing thirteen simulations in which these five providers participated, and showed no statistically significant difference in the performances under each study arm.

Eleven pairs in every 3-minute rotation and eight pairs in every 5- and 10- minute rotation schedules performed CCs with an adequate depth greater than 90% of the session duration. Two pairs in every 3- and 5- minute rotation schedules and four pairs in every 10-minute rotation schedule performed CC at an adequate depth less than 10% of the session duration. The composition of each of these pairing was different for the three simulation types.

Regardless of the lack of any evident decay in overall CC performance as providers administered CCs in the three schedules, providers reported a significantly higher level of fatigue while performing CCs for longer durations, as shown in the last box-plot in Fig 3. The number of CCs missed during the time of rotation are reported in Table 4. More CCs were missed when providers rotated out more frequently, however, the number of CCs missed were not statistically significant when compared to the total number of CCs performed over the 18 and 20 minute runs. In S1 Table, we present the median differences between the pre-simulation values and values obtained after each round of CCs in the providers' blood pressure, heart rate and oxygen saturation, removing the outliers.

## Discussion

In this simulated randomized controlled study, we did not find any statistically significant difference in CC performance as providers rotated more frequently. However, we observed a significant increase in the providers' self-reported fatigue scores and significant rise in providers' heart rate and systolic blood pressures when performing CCs for a longer duration. Similar findings were reported in an adult simulation study where paired performance of continuous CCs for 1 vs. 2 minutes yielded no significant difference in the effective CCs, however, there was substantially more fatigue reported by providers rotating every 2-minutes as opposed to every 1-minute [13].

Unlike one previous study, in which a 20% reduction in CC depth was reported after three minutes of 3:1 CV compressions, we found no statistically significant deterioration in compression depths as providers rotated more frequently [9]. This finding could result from the differences in the technique and type of manikin used in the two studies. CCs were performed

**Table 3. Characteristics of the participants.**

| Variables | N | Median (25th%-75th%) | Mean (SD) |
|---|---|---|---|
| Age (years) | 51 | 37.0 (33.0,43.0) | 39.8 (9.92) |
| Weight (kg) | 50 | 75.0 (65.0, 93.0) | 79.7 (19.9) |
| Height (inch) | 51 | 65.0 (62.0, 68.0) | 65.9 (6.19) |
| Hours of Physical Activity | 51 | 2.00 (0.00, 3.00) | 2.47 (2.66) |

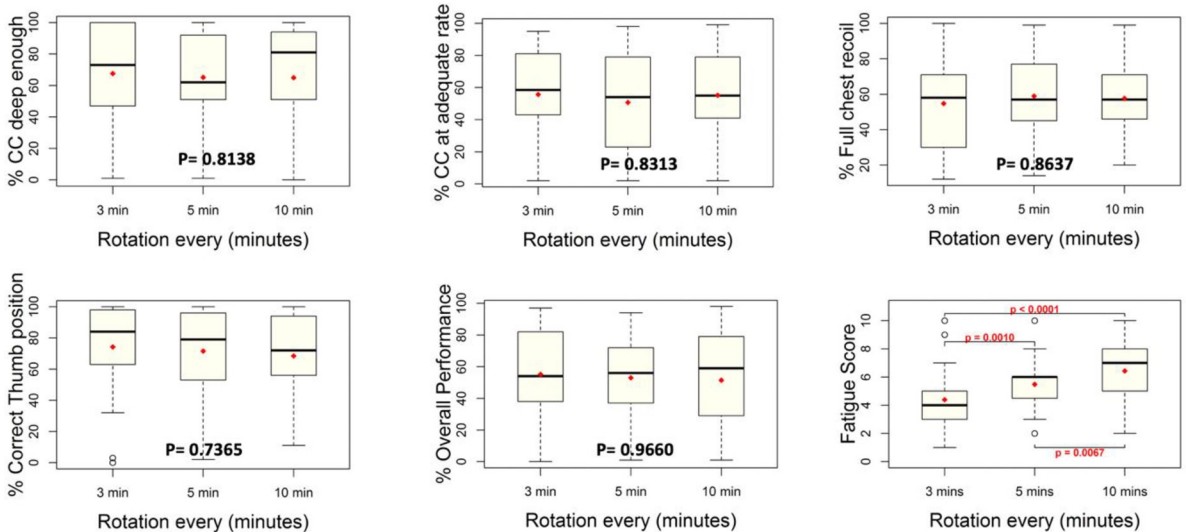

**Fig 3. Box-plots comparing different compression performance metrics and provider self-reported fatigue scores, as they rotated every 3, 5 and 10 minutes.** Statistically significant difference was noted in the level of fatigue when providers performed CCs for longer durations Note: p-values reported within each box-plot.

from the head of the bed in our study and from the side of the bed in the other study. The variation in manikin size and chest compliance can also have a significant impact on the work required to perform compressions.

In a previous study by our group, we similarly reported a decline in performance quality as early as 3.5 minutes from the start of CCs [10]. The same manikin and skills reporting device were used in both the studies. However, in the previous study, CC depth of one-third of chest diameter was defined as adequate quality, and a surrogate marker for provider fatigue was defined as four consecutive CCs below target depth. They did not analyse any other compression metrics. The previous study made use of the HeartStart Mrx monitor which allowed for quantification of the rate, depth, chest deflection and residual leaning force. The current study solely used the Resusci Anne Skill Reporter device that provided composite data on the entire run, which included the average CC depth and % CC that were deep enough. The primary goal for the previous study was to assess the impact of fatigue on the quality of CCs by demonstrating weariness, which was present in 52% of the providers. This finding led us to test our current hypothesis that frequent rotation would improve the providers' performance and lower their level of fatigue during a prolonged neonatal resuscitation.

Contrasting our findings with those of a study using paediatric and adult manikins where a trend of increasing CC rate as the CC depth decreased overtime during a 10-minute resuscitation was reported, we observed neither a significant increase in CC rate nor a decrease in CC depth in scenarios that required less frequent rotations [4]. Two other adult studies, in contrast, reported a trend of declining CC rate overtime alongside the reduction in performance [14,15].

**Table 4. Calculated number of missed CCs based on the duration of pauses and average CC rate for the simulations.**

| Rotation Frequency | Mean CC rate (SD) | Sum of mean pause duration (seconds) | Calculated # of CCs missed |
|---|---|---|---|
| 3 minutes | 112 (9.45) | 19.89 | 37.13 |
| 5 minutes | 116 (14.7) | 12.30 | 23.78 |
| 10 minutes | 115 (11.9) | 4.44 | 8.51 |

These disparities could be due to the major difference in the sizes of paediatric and adult manikins compared with neonatal manikins, and the provider position relative to the manikin, along with the recommended hand technique used in BLS and PALS versus NRP and ERC.

Despite the demonstration of an ideal CC quality and practice time allowed prior to the simulation session, five providers rarely administered CCs of an adequate depth. It is uncertain whether this finding is due to their poorer baseline skills or to their tendency to approach the simulation scenario in a relatively casual manner. We found providers who consistently performed CCs at an adequate depth >90% and for <10% of the time regardless of the rotation frequency. This finding suggests the possibility of a significant amount of inherent variability in the ability of trained providers to perform adequate CCs. A similar observation was made in an adult simulation study in which 47% of providers performed poor CCs from the initiation to the end, suggesting that the variability in the providers' performance was more likely associated with the poor CC quality than with fatigue [16]. Of note, the skills reporting device used in the present study does not report over compression, i.e. CCs that were performed at a depth greater than what is considered adequate (1/3 AP diameter), hence it is hard to speculate if providers that compressed deeper had higher fatigue. A more recent study attempted to eliminate this confounding factor by demonstrating adequate CC quality in the 30-second practice sessions before initiation of the session and yet noted decay (albeit at different rates) for all participants during each session [4]. We allowed a similar practice time and feedback before the initiation of the study session. Performances during practice were not recorded and no verbal feedback was shared, but the participants were able to visually track the compression depth and hand positioning on the SimPad PLUS with Skill reporter device during this practice period.

The observed increase in the providers' heart rate and systolic blood pressure over time was most likely attributable to the effort required to perform CCs. As expected, the change from baseline is substantial for simulations with less frequent rotation. This occurrence is well documented in many adults, pediatrics and neonatal studies [10,17,18].

## Limitations

Simulation study can never capture all elements of a real-life scenario, despite the best measures taken to mimic one. The psychological stress and the resulting adrenaline surge that providers experience from a real code situation are impossible to replicate in a simulated scenario. Providing live feedback in efforts to mimic a real-world in-hospital code could have significantly improved the CC quality of some of the persistently poor performers, however this would have violated the integrity of the study protocol. Technological limitations in extracting data for the performance of individual providers made it impossible to make a true comparison of an individual provider performance under the three simulation models, as well as a comparison of their performance from the first round of CCs to the next. Lastly, the questionnaire used in this study was not validated, leaving a possibility of measurement error.

## Future directions

Ideally, we would like to replicate this study once we are able to obtain a software with the ability to breakdown the simulation data in brief windows of time, allowing us to evaluate the decay in performance overtime in relation to the last rotation, as well as individual provider performance over the three simulations.

## Conclusions

In conclusion, rotating every 3, 5 or 10 minutes did not affect any of the CC performance metrics. However, providers' self-reported fatigue scores, heart rates and systolic blood pressures

significantly increased with longer CC durations. Our findings support that the current practice of rotating providers when feeling fatigued is appropriate during a prolonged neonatal CPR. Furthermore, given the infrequent requirement of chest compressions in neonates, there might be some utility in incorporating a periodical assessment of the providers CC performance quality.

## Supporting information

**S1 Fig. Participant questionnaire.**
(DOCX)

**S1 Table.**
(DOCX)

## Acknowledgments

We would like to thank all the study participants for the enthusiasm with which they participated in this study, the research staff for all their contribution, and Dr. Anne Wlodaver for providing us with her unbiased and critical feedback.

## Author Contributions

**Conceptualization:** Tavleen Sandhu, Edgardo G. Szyld, Birju A. Shah.

**Data curation:** Tavleen Sandhu, Michael P. Anderson.

**Formal analysis:** Michael P. Anderson.

**Investigation:** Tavleen Sandhu, Edgardo G. Szyld, Birju A. Shah.

**Methodology:** Tavleen Sandhu, Edgardo G. Szyld, Michael P. Anderson, Birju A. Shah.

**Project administration:** Tavleen Sandhu.

**Supervision:** Tavleen Sandhu, Edgardo G. Szyld, Birju A. Shah.

**Validation:** Tavleen Sandhu, Edgardo G. Szyld, Michael P. Anderson, Birju A. Shah.

**Visualization:** Tavleen Sandhu.

**Writing – original draft:** Tavleen Sandhu.

**Writing – review & editing:** Tavleen Sandhu, Edgardo G. Szyld, Michael P. Anderson, Birju A. Shah.

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
