## [Decision Letter · Decision Letter 0]

30 Dec 2021

PONE-D-21-22882Effect of rotating providers on chest compression performance during simulated neonatal cardiopulmonary resuscitationPLOS ONE

Dear Dr. Sandhu,

Thank you for submitting your manuscript to PLOS ONE. After careful consideration, we feel that it has merit but does not fully meet PLOS ONE’s publication criteria as it currently stands. Therefore, we invite you to submit a revised version of the manuscript that addresses the points raised during the review process.

Cover letter is addressed to Journal of Perinatology and Patrick Gallagher. Would advise authors to carefully read materials submitted to ensure correct information.

We look forward to receiving your revised manuscript.

Kind regards,

Jayasree Nair, MBBS MD FAAP

Academic Editor

PLOS ONE

Journal Requirements:

3. Please include additional information regarding the survey or questionnaire used in the study and ensure that you have provided sufficient details that others could replicate the analyses. For instance, if you developed a questionnaire as part of this study and it is not under a copyright more restrictive than CC-BY, please include a copy, in both the original language and English, as Supporting Information. If the original language is written in non-Latin characters, for example Amharic, Chinese, or Korean, please use a file format that ensures these characters are visible.

4. Please state whether you validated the questionnaire prior to testing on study participants. Please provide details regarding the validation group within the methods section.

Additional Editor Comments:Positives: authors have analyzed physiological data of fatigueMinor revisions: 75 sessions line 106. 76 in table. Please ensure consistencyTable 4 headings are missing- table is not explained properly. It seems more CC are “missed” when providers are rotated out- this seems significant?Page 12: “However, in the previous study, CC depth of one-third of chest diameter was defined as adequate quality, and a surrogate marker for provider fatigue was defined as four consecutive CCs below target depth.” This data is available for current study as well. Have the authors looked at these? From fig 2- it looked like CC depth was similar between groups here? How do you explain different findings/interpretation in same mannequin and relatively similar group of participants? If you are showing increased fatigue in 10min- why do you not see effect on quality/depth of CC compared to shorter durations?Page 12 line 238: you report under compression by providers. Was there “over compression” ie compressing excessive depth by some providers- that could impact fatigue.

Reviewers' comments:

Reviewer's Responses to Questions

**Comments to the Author**

1. Is the manuscript technically sound, and do the data support the conclusions?

Reviewer #1: Yes

2. Has the statistical analysis been performed appropriately and rigorously? 

Reviewer #1: Yes

3. Have the authors made all data underlying the findings in their manuscript fully available?

Reviewer #1: Yes

4. Is the manuscript presented in an intelligible fashion and written in standard English?

Reviewer #1: Yes

5. Review Comments to the Author

Reviewer #1: In this study, the authors present interesting data on the effect of chest compression (CC) rotation on CC performance and provider fatigue in a simulation manikin study.

The authors studied 51 providers paired to deliver CC at 3 min, 5 min and 10 min intervals for a duration of 18 – 20 min. There was no deterioration in CC performance (percent CCs with adequate rate, adequate depth, complete recoil, correct thumb position, and duration of overall pause), but providers reported higher fatigue with longer CC intervals. Provider fatigue was assessed by heart rate and blood pressure measurement, as well as reporting level of fatigue on a scale of 1-10. Five providers consistently had poor CC performance with depth measured at < 10%. Data were re-analysed after removing thirteen simulations in which these five providers participated, and showed no statistically significant difference in the performances under each study arm.

This reviewer has the following minor comments/suggestions:

1. p3, line 63: "bpm" should be spelled out the first time the abbreviation is used.

2. p3: ref 7 and 8 should be updated to the more recent 2020 publications.

3. page 3, line 65: "bpm" abbreviation can be used here.

4. p5, line 94: CV should be spelled out as this is the first time the abbreviation has been used.

5. p11, Table 4: Missing a title row for the respective columns (e.g. what do the 19.89 and 37.13 values for the 3 min row indicate?).

6. Methods/Results: The authors mention in the methods section that reporter fatigue was assessed by Likert scale. Can the authors include those results in the manuscript?

7. Discussion, p13: can the authors comment if the five providers who performed chest compressions at <10% depth had a similar performance during their practice time and how they responded to the feedback? One would expect their performance to have improved during the simulation following the practice session.

6. PLOS authors have the option to publish the peer review history of their article (what does this mean?). If published, this will include your full peer review and any attached files.

Reviewer #1: No

---

## [Author Response · Author response to Decision Letter 0]

14 Jan 2022

Response to the editors’ comments:

• Minor revisions: 75 sessions line 106. 76 in table. Please ensure consistency.

o Line 106 mentions what was intended in the protocol, however due to error in pairing of the participants we were left with less than 75 sessions and hence enrolled one additional participant and ended up with a total of 76 sessions (this is explained in the manuscript lines 192-195). Additionally, we have added this information in the text under figure 2 to provide more clarity.

• Table 4 headings are missing- table is not explained properly. It seems more CC are “missed” when providers are rotated out- this seems significant?

o The table heading has been addressed, sorry for that miss earlier

o It is true that the more CC were missed when the providers rotated out more frequently, however the number of CCs missed were not statistically significant when compared to the total number of the CC performed over the 18 and 20 minute runs. We have elaborated on this now in lines 214-217 for additional clarity.

• Page 12: “However, in the previous study, CC depth of one-third of chest diameter was defined as adequate quality, and a surrogate marker for provider fatigue was defined as four consecutive CCs below target depth.” This data is available for current study as well. Have the authors looked at these? From fig 2- it looked like CC depth was similar between groups here? How do you explain different findings/interpretation in same mannequin and relatively similar group of participants? If you are showing increased fatigue in 10min- why do you not see effect on quality/depth of CC compared to shorter durations?

o The previous study had used the HeartStart Mrx monitor which allowed quantification of the rate, depth, chest deflection and residual leaning force. We solely used the Resusci Anne Skill Reporter device which provided composite data on the entire run, this included the average CC depth and % CC that were deep enough. We have included this piece of information now in line 255-258. 

• Page 12 line 238: you report under compression by providers. Was there “over compression” ie compressing excessive depth by some providers- that could impact fatigue.

o Over compression was not reported by the device. It only reports % compressions that were of adequate depth (>1/3 AP diameter). We have elaborated on this in lines 288-291.

Response to reviewers’ comments:

1. p3, line 63: "bpm" should be spelled out the first time the abbreviation is used.

- Done

2. p3: ref 7 and 8 should be updated to the more recent 2020 publications.

- Done

3. page 3, line 65: "bpm" abbreviation can be used here.

- Done

4. p5, line 94: CV should be spelled out as this is the first time the abbreviation has been used.

- Thank you, we have now addressed that in line 67.

5. p11, Table 4: Missing a title row for the respective columns (e.g. what do the 19.89 and 37.13 values for the 3 min row indicate?).

- Done, apologies for that missing information. 

6. Methods/Results: The authors mention in the methods section that reporter fatigue was assessed by Likert scale. Can the authors include those results in the manuscript?

- This is shared in the last box plot on the Figure 3 (see below for reference) and mentioned in lines 211-213. Additionally, we have now included this information in the text under the figure. 

7. Discussion, p13: can the authors comment if the five providers who performed chest compressions at <10% depth had a similar performance during their practice time and how they responded to the feedback? One would expect their performance to have improved during the simulation following the practice session.

-Performances during practice were not recorded. No verbal feedback was shared during practice either, but the participants were able to visually track the compression depth and hand positioning on the SimPad PLUS with Skill reporter device during this practice period. I have included this piece of information in lines 295-298.

We are hopeful that the revisions that have been made have successfully addressed the points that were made by the editor and reviewer. However, if any further clarification is needed, we will be happy to address that. We thank the team of PLOS ONE once again for considering this manuscript for publication.

---

## [Editor Report · Decision Letter 1]

23 Feb 2022

Effect of rotating providers on chest compression performance during simulated neonatal cardiopulmonary resuscitation

PONE-D-21-22882R1

Dear Dr. Sandhu,

We’re pleased to inform you that your manuscript has been judged scientifically suitable for publication and will be formally accepted for publication once it meets all outstanding technical requirements.

Kind regards,

Jayasree Nair, MBBS MD FAAP

Academic Editor

PLOS ONE
---

## [Editor Report · Acceptance letter]

28 Feb 2022

PONE-D-21-22882R1 

Effect of rotating providers on chest compression performance during simulated neonatal cardiopulmonary resuscitation 

Dear Dr. Sandhu:

I'm pleased to inform you that your manuscript has been deemed suitable for publication in PLOS ONE. Congratulations! Your manuscript is now with our production department. 

Kind regards, 

on behalf of

Dr. Jayasree Nair 

Academic Editor

PLOS ONE